# An integrated model of the younger generation's online shopping behavior based on empirical evidence gathered from an emerging economy

**Van Thac Dang**[1], **Jianming Wang**[2]*, **Thinh Truong Vu**[3]

**1** Department of Business Administration, Business School, Shantou University, Shantou, Guangdong, China, **2** School of Business Administration, Zhejiang University of Finance & Economics, Hangzhou, China, **3** Department of Business Administration, Dong Nai Technology University, Bien Hoa, Vietnam

* sjwjm@zufe.edu.cn

## Abstract

The younger generation is the largest Internet user group in China. They are the first generation to grow up with computers, the Internet, smartphones, online social media, and online shopping. The individuals that belong to this generational cohort have one thing in common —their online shopping behavior. To understand the shopping behavior of the younger Chinese generation, this study draws on the theoretical foundation of generational cohort theory. This study proposes an integrated model that examines the effects of information adoption, personalized service, perceived switching risk, and habitual behavior on purchase intention in the online shopping environment. Survey data have been collected from 407 Chinese people that belong to the post-90s generation. Structural equation modeling is used to analyze the data. Empirical findings show that information adoption, personalized service, and perceived switching risk are the most important predictors of online purchase intention. However, habitual behavior is negatively related to online purchase intention.

## 1. Introduction

Recent Internet technologies have provided an important vehicle for commercial transaction around the world. Internet commerce refers to the transaction of products and services via the Internet [1]. Internet commerce enables simple, easy, and cheap methods for consumers and business firms to interact and conduct business electronically [2]. The Internet is altering the nature of customer shopping behavior and our society [3]. Furthermore, China has become the world's second largest economy and the biggest Internet market [4]. According to the China E-commerce Research Center, Internet users in China have increased from 731 million in 2016 to 772 million in 2017, thereby accounting for over 59.4% of its population. Chinese online shopping users have reached 533 million in 2017 and 560 million in 2018. In addition, online retail sales have increased to nearly 1.1 trillion US dollars in 2017, accounting for 19.6% of its total retail sales. These retail sales are expected to reach nearly 1.5 trillion US dollars in

**Data Availability Statement:** Data cannot be shared publicly because they contain potentially sensitive information and will provide analysis for third-party company, DaXue Marketing Research

and Management Consultant Company. Data were collected upon the approval of Shantou University Business School Ethics Committee. Data are a part of the project between Shantou Business school and the DaXue Marketing Research and Management Consultant Company (contact via Manager Dr. Vi Thanh, greenstardeng84@yahoo. com). The authors of the present study had no special access privileges in accessing data from this third-party that other interested researchers would not have. The data set (entitled "Chinese Younger Generation Online Shopping Behavior") is available from the Shantou University Business School Ethics Committee (contact via Administrator Dr. Wilson Dang, wilsondang1005@gmail.com).

**Funding:** This study is funded by the Guangdong Higher Education Major Research Project (grand number: 2018WQNCX036) and the National Natural Science Foundation of China (Grant No.71673238).

**Competing interests:** The authors have declared that no competing interests exist.

2018. E-commerce B2C is expected to be the fastest growing segment of the retail industry in China. Therefore, e-commerce B2C holds its high promises for both academic researchers and business managers to explore the high-potential e-commerce market in China.

A growing number of research has started to examine China's e-commerce market [5–13]. However, prior studies on China's e-commerce market tend to treat China consumers as an ethnically homogenous group [14]. Bilgihan [15] has stated that generational cohorts have different values, attitudes, preferences, and shopping behaviors. Business managers must understand such differences and provide suitable products and services for different consumer groups. According to the China Internet Network Information Center, Chinese internet users, aged 20 to 29 years old, account for 30% of China's population. These are the largest Internet users group in China. As reported in mass media, the younger Chinese generation takes smartphones, tablets, wireless Internet, and digital media for granted, blending the online worlds seamlessly as they socialize and shop [16]. The gap between China's new and older generations is arguably wider than that in Western societies due to the dramatic transformation in China's economic, politics, and culture in the last decades [17]. Despite the difference between Chinese generations, a lack of social scientific research on the younger Chinese generation remains. Therefore, a comprehensive model describing the factors that influence the online shopping behavior of the younger Chinese generation will offer insights to academics and practitioners.

Generational difference has become an important gap since social value changed through generation replacement in China [18]. Transitional economy like China has witnessed a wider gap between generations after its social reform [19]. Although generational cohort theory is universally applicable in Western societies [15], an investigation of generational value differences in China would shed a new light on the generalization of the theory. Furthermore, prior researchers tend to apply generational cohort theory in the fields of organizational behavior and human resource management [17, 20]. The absence of its application in marketing management limits the generalizability of the theory. Investigating online shopping behavior of Chinese younger generation would extend generational cohort theory in the field of marketing and consumer behavior (i.e. e-commerce). In addition, a lack of empirical evidence on consumer behavior of Chinese younger generation has provided very limited knowledge to business managers' decision making in China market. Findings of this study will contribute to enhance quality of business policy for Internet commerce in China. Therefore, this study draws on the theoretical foundation of generational cohort theory to investigate important factors that affect the online shopping behavior of the post-90s Chinese generation.

In the context of e-commerce, consumers often collect numerous information to reduce product and purchase uncertainty in their decision-making process. Information adoption can be viewed as an important factor to consumer purchase behavior because information helps consumers evaluate and understand products, services, and everything related to their purchase decisions [11]. Moreover, the absence of face-to-face interaction with consumers leads to the recognition of the inadequacy of traditional methods of service. Personalized service becomes particularly important for online stores [21]. Online retailers, such as Amazon and Jingdong.com (China), have been successful because they provide excellent, efficient, responsive, flexible, guaranteed, interactive, supportive, and informative customized service [22]. Personalized service is identified as a key factor that influences consumer behavior in the online shopping environment [21, 23]. Furthermore, in an era of rapidly changing technology and highly complex markets, consumers often have too many choices for their purchase. A diversified market with highly homogeneous products makes it easier and cheaper for customers to switch between retailers [24]. Switching risk becomes an important factor that determines consumers' attachment and loyalty to a particular online retailer [25]. Compared with physical stores, online retailers offer an extensive selection of products, timely information,

and added personalized information about the products. Online shopping also offers convenience, time saving, and low transaction costs for consumers and business firms [3]. Such tremendous benefits make consumers become accustomed to online shopping. The role of habitual behavior may also play an important role in the consumer decision-making process [26]. To advance our knowledge on the post-90s Chinese generation's online purchase behavior, this study proposes an integrated model. This study also investigates the influence of information adoption, personalized service, perceived switching risk, and habitual behavior on purchase intention of the generational cohort born after 1990 in the online environment of China.

Following this introduction, Section 2 presents the theoretical background and research hypotheses. Section 3 covers the study methodology. Section 4 presents the results and findings. Section 5 discusses the conclusions and limitations and proposes directions for future research.

## 2. Theoretical background and hypotheses

### 2.1. Generational cohort theory and Chinese generational cohort

Generational cohort theory posits that people can be classified into different groups or generations based on the era they were born [18]. People who were born at the same time and space tend to share similar values, beliefs, and life experiences. These similarities influence people's attitudes, preferences, and behaviors [27]. Different generations have different characteristics, expectations, and views due to their exposure to various social contexts [15]. Thus, a generational cohort is defined as a group of people who were born at approximately the same time and experienced similar social and life events that shaped their unique values, attitudes, and behaviors [28].

Studies on Chinese generational cohort have divided generations based on key historical events. Egri and Ralston [29] and Wang ([30], in Chinese) have used a sample data of employees and the general public to divide people into the Republican Era (born between 1930 and 1950), the Consolidation Era (born between 1951 and 1960), the Cultural Revolution Era (born between 1961 and 1970), and the Social Reform Era (born between 1971 and 1975). Most scholars have classified Chinese generations by decade. Liu ([20], in Chinese) and Tang et al. [17] have used employees as sample data and classified them into pre-reform generation (born before 1978), reform generation (born between 1979 and 1989), and post-reform generation (born after 1990). The most recent accepted categorization of Chinese generations is the cohorts known as the post-70s generation (born between 1970 and 1979), post-80s generation (born between 1980 and 1989), post-90s generation (born between 1990 and 1999), and post-00's generation (born after 2000) [19].

The Chinese post-90s generation was born in an era of the Internet and digital technology. They were the first generation to grow up with computers, internet, mobile phones, online social media, and online shopping. After Jack Ma launched Alibaba.com in 1998 [31], the Chinese post-90s generation has been accompanied with the development and growth of e-commerce market in China. The members of this generational cohort have one thing in common —their comfort with online shopping. They view online shopping as an indispensable activity in their daily lives. They also have a unique attitude and value toward their shopping behaviors [17]. Therefore, e-commerce marketers should understand this generational cohort and use suitable strategies to provide products and services for them. The next section discusses the influential factors that affect the online purchasing behavior of the post-90s Chinese generation.

## 2.2. Information adoption theory and technology acceptance model

Information adoption theory aims to explain the important factors that drive people to adopt information or technology in an organizational context [32]. Information adoption theory is rooted in the theory of reasoned action [33] and the technology acceptance model (TAM) [34]. These models suggest that beliefs are important determinants of individuals' intention to adopt particular information or technology. These beliefs may include perceived ease of use and perceived usefulness. The former refers to "the degree to which an individual believes that using a particular technology would be free of physical and mental effort." The latter refers to "the degree to which an individual believes that using a particular technology would enhance his or her job performance" [33]. Information adoption theory has been applied by several researchers to explain the adoption of new technology, such as software products [35, 36], spreadsheets [37], voice mail [38], and information systems [11,36]. In this study, information adoption theory and TAM are applied to explain information factors (e.g. source credibility, information quality, information usefulness, and information adoption) that affect Chinese younger generation's shopping behavior in online environment.

## 2.3. Information adoption in online shopping

According to information adoption theory and TAM, the usefulness of information plays a critical role in the consumer decision-making process [39]. Sussman and Siegal [32] have argued that source credibility and information quality are two major factors that influence information usefulness. Source credibility refers to the reliability of information sources rather than the content of the information. Information quality refers to the completion, consistency, and accuracy of the information [40]. In the context of online shopping, source credibility explains that the people who provide information or comments on shopping websites are knowledgeable, reliable, and trustworthy [11]. Consumers tend to trust information from credible sources, and they often feel that source credibility is useful because it is beneficial for their purchase decisions [41]. Consequently, consumers will perceive information provided by credible people as useful information. Thus, the following hypothesis is developed.

*H1. Source credibility is positively related to information usefulness*

Consumers often perceive a high risk in online shopping because of the lack of face-to-face communication [42]. For example, consumers may worry that the product is counterfeit, of low-quality, or that the quality of the product is different from that of the product they viewed on the screen [25, 43]. Consequently, information quality is useful for consumers to evaluate and assess products and services online [32]. Consumers are more likely to perceive that high information quality is useful because it enhances the quality of consumer purchase decision [11]. Thus, the following hypothesis is developed.

*H2. Information quality is positively related to information usefulness*

In the Internet age, consumers often search numerous information from different sources before purchasing a product. When consumers lack information or obtain low-quality information, they often face higher uncertainty in their decision-making process [25]. By contrast, if consumers obtain useful information, they will better understand products, services, and other policies related to their purchase decisions [43, 44]. The influence of information usefulness on information adoption has been confirmed in prior studies [11, 39]. Based on the above argument and prior research, the following hypothesis is developed.

*H3. Information usefulness is positively related to information adoption*

Kotler and Armstrong [45] have argued that information search is an important step in the consumer purchase decision-making process. Seeking information helps consumers find out useful information about products. Obtaining useful information reduces information

asymmetry and costs related to their decisions [46]. Furthermore, consumers adopt useful information to enhance their understanding about competing brands and their features and to advance consumer knowledge and insight about a particular product or service [41]. As a result, information adoption helps consumers reach a better purchase decision. Thus, the following hypothesis is developed.

*H4. Information adoption is positively related to online purchase intention*

## 2.4. Personalized service in online shopping

Personalized service is a process whereby products and services are tailored to match individual consumer needs and demands [47]. This service refers to the interaction between an online retailer and a customer on a one-to-one basis [48]. In the context of online shopping, personalized service is often provided in the form of personalized information, suggestions, recommendations, and multiple versions of interaction touchpoints [49]. Online retailers, such as Amazon, eBay, Taobao, and Apple, use personalized services to interact with their consumers. They allow consumers to use the embedded toolbars in their website's portal and filter information related to the consumers' personalized demand [50]. Furthermore, online retailers often use personalization as an effective strategy to persuade consumers to select and purchase a product or service [48]. Examples of personalization services include voice interaction or live chat between service employees and online shoppers, rapid responses to consumers' demands, prompt delivery of products, ease of payment, ease of product return, specific services for individual consumer's demands, and website design personalization [47, 51]. In a rapidly changing and intensely competitive environment, online retailers depend heavily on personalized services to fulfill consumer needs. One-to-one marketing helps online retailers improve their understanding of their consumers. Online retailers can also learn consumer preferences and synthesize the gathered knowledge from the personalization process into their products and services [48]. Consequently, personalized services have become an important source of competitive advantage for online retailers [3].

Levy and Weitz [3] have stated that online retailers create great benefits for consumers because they personalize merchandize offerings and information for each consumer. Consumers can obtain tremendous information to make a purchase decision or they can format the information to compare the product with other alternatives. Given the interactive nature of the personalization process, personalization raises emotional and rational assurance of consumer choices, increases consumer trust in online retailers, and enhances consumer satisfaction [52]. Personalization gives online shoppers pleasant shopping experiences that are previously available only in the physical shopping environment [53]. Personalized services make online shopping more interesting, more hedonic, and more economical for consumers [54]. As personalized service enhances the consumers' emotional and economic benefits when shopping online [3, 53], consumer intention to purchase in online increases. Thus, the following hypothesis is developed.

*H5. Personalized service is positively related to online purchase intention*

According to behavioral learning theory, people often learn to perform behaviors that produce positive outcomes and avoid those that yield negative results [55,56]. A high level of personalization implies that consumers can enjoy more values and positive rewards from online retailers. For example, when consumers purchase personalized products that fit their specific demands, they can obtain high-quality services [57]. Consumers can also gain benefits from a customized combination of information, product, promotion, price, delivery, and service [58]. When consumers enjoy high-value personalized service, they tend to hold positive attitudes and trust the retailer; in addition, emotional and behavioral dependence are likely to emerge from the personalization process [54]. In other words, personalized service acts as an

instrumental mechanism that induces consumer attitudes and behaviors. When consumers gain positive outcomes from the personalization process, they hold positive attitudes toward the online retailer. Over a long period and repeated process, consumers become familiar and emotionally attached to the online retailer. Consequently, personalization process leads to consumer habitual behavior because consumers depend on the personalized service of online retailers to achieve positive outcomes. Thus, the following hypothesis is developed.

*H6. Personalized service is positively related to habitual behavior*

In addition, a high level of personalized service makes it harder for consumers to switch to alternative retailers because personalization raises consumer perceptions of switching risk [59]. Online retailers, such as Amazon and Jingdong.com, often create personalized webpages that allow each user to generate and maintain his or her own profile page. This personalization service contributes to the creation of personal assets and reduces transaction costs for consumers [60, 61]. Furthermore, the decision-making process requires a certain amount of time, energy, and effort for consumers to be familiar with a specific online retailer. A high-level of familiarity implies that consumers obtain additional knowledge and experience from that online store. Thus, personalization makes consumers stay with familiar online stores because doing so costs less time, energy, effort, and money [59]. By contrast, if consumers switch to a new retailer, they may face higher levels of risk and uncertainty. For example, consumers may not enjoy the personalized services as much as those provided by the current online retailer; they are also unsure of the reliability and security of the new retailer; in addition, they may have to spend much more time and effort to understand the new retailer [53]. Therefore, personalization raises consumer perceptions of risk when they intend to switch to other retailers. The following hypothesis is developed.

*H7. Personalized service is positively related to perceived switching risk*

## 2.5. Perceived switching risk in online shopping

Protection motivation theory posits that threat appraisal and coping appraisal are cognitive processes aroused when an individual faces a threat, which motivates protective behavior to reduce or avoid the threat [62, 63]. Threat appraisal is determined by perceived severity and perceived susceptibility to risks. Coping appraisal is associated with self-efficacy and response efficacy [64]. Self-efficacy is the belief that individuals can successfully perform protective behavior. Response efficacy believes in the effectiveness of the protection [65]. Protection motivation theory has been used to explain and understand protection behaviors online [64, 65]; [66].

Internet shoppers often feel a high risk in online activities [65]. Several risks and threats are associated with online shoppers, such as identity theft, malware or viruses, security of financial information, and information privacy [67]. When switching to new retailers, consumers often face high risks and threats because they are uncertain about the new retailer. Consequently, they may have to spend much more time and energy to become familiar with the new retailers. According to protection motivation theory, consumers tend to perform protective behaviors to reduce risks and threats when they perceive and evaluate that switching risks are high and severe, [62, 63]. Choosing to purchase from a specific store will be effective if the current online retailer provides reliability and high-quality products and services and if consumers are familiar with that retailer [43, 68]. By contrast, when switching costs are low, consumers may try to purchase from new retailers because they can have a novel shopping experience at lower costs. Such switching behavior enhances consumers' hedonic experience and satisfaction [3]. Thus, consumer perceptions of switching risk are expected to enhance consumers' intention to purchase from current online stores. The following hypothesis is developed.

*H8. Perceived switching risk is positively related to online purchase intention*

## 2.6. Habitual behavior in online shopping

Habits are defined as "routine behaviors that repeat regularly and tend to occur subconsciously" [69]. Habits form as people repeatedly and consistently perform a particular behavior to meet a goal [70]. Prior studies have claimed that approximately 45% of people's behavior is repeated almost daily and eventually turns into habitual behavior [26]. Habits have received much attention in prior studies, probably because numerous purchase and consumption activities are associated with habits [71]. For example, Seetharaman [72] has found that consumers tend to repeatedly buy the same brands of products at different shopping outlets. Khare and Inman [73] have indicated that people eat similar types of foods for days. Khalifa and Liu [74] have demonstrated that online shopping habits can positively predict online shopping satisfaction. Hsiao, Chang, and Tang [75] have suggested that habit is an important antecedent of continuous intention. Similarly, habitual purchase behavior is also an important phenomenon in the online shopping environment [71].

Habitual behavior in the online environment can be viewed as an instrumental learning process [70]. According to instrumental learning theory, consumers learn to associate positive rewards with purchase behavior when shopping online [76]. Stimuli, such as customized information and recommendations, high-quality products, low prices, and a personalized webpage, often act as rewards that induce consumer purchase behavior. If this process repeats over time, consumers will learn to link these stimuli with benefits and values when purchasing online [71]. Consequently, online purchasing behavior is strengthened and becomes habitual behavior because consumers achieve positive outcomes each time they shop online. Accordingly, this study proposes the following hypothesis (Fig 1 shows the research framework of this study).

*H9. Habitual behavior is positively related to online purchase intention*

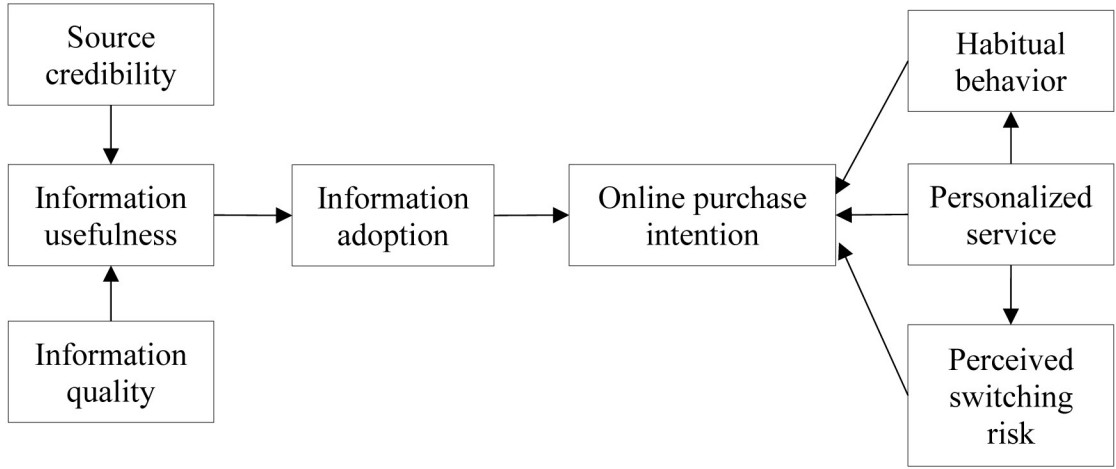

Figure 1. Research framework

**Fig 1. Research framework.**

## 3. Research methodology

### 3.1. Sample and data collection

The data were collected from four large different universities in China. We obtained approval from the Business School's Ethical Committee of Shantou University to conduct the survey. The researchers randomly invited college students to shop on Taobao.com and accomplish the survey. The students were born after 1990, thus representing the post-90s Chinese generation. These students voluntarily participated in the survey. Our research team explained the purpose and content of the survey before the respondents completed the questionnaire. The questionnaire was anonymous. Therefore, respondents could provide their measures without privacy concerns. The volunteer sampling technique was used because of the absence of a proper sampling frame in this study. After 4 months, this study finally obtained 407 completed surveys for the data analysis.

Table 1 shows the demographic profile of the sample. The respondents comprised 116 males (28.5%) and 291 females (71.5%). Approximately 84.8% of the respondents were freshmen, 9.6% respondents were sophomores, 3.7% respondents were juniors, and the remaining 2.0% respondents were seniors. Approximately 71.0% of the respondents were 20 years old or below, 26.5% were between 21 and 24 years old, and only 2.5% were between 24 and 28 years old. About 96.6% of the respondents purchased online more than three times a month, and only 3.4% of the respondents purchased online less than three times a month.

### 3.2. Instrumentation

A preliminary version of the questionnaire was formulated using forward and backward translation from English to Chinese, and vice versa. Two professional translators who are proficient in both English and Chinese were recruited to perform the forward and backward translation. A pilot test was conducted with 26 students in the author's university to check the appropriateness of the wording and meaning of items in the questionnaire. The survey questionnaire was

**Table 1. Demographic profile.**

| Variable | Sample | Percentage (%) |
|---|---|---|
| Gender | | |
| *Male* | 116 | 28.5% |
| *Female* | 291 | 71.5% |
| Education | | |
| *Freshman* | 345 | 84.8% |
| *Sophomore* | 39 | 9.6% |
| *Junior* | 15 | 3.7% |
| *Senior* | 8 | 2.0% |
| Age | | |
| *Under 20* | 289 | 71.0% |
| *21–24* | 108 | 26.5% |
| *25–28* | 10 | 2.5% |
| *28 or above* | 0 | 0.0% |
| Online shopping frequency (times/per month) | | |
| *Not at all* | 0 | 0.0% |
| *1* | 0 | 0.0% |
| *2* | 14 | 3.4% |
| *3 or above* | 393 | 96.6% |

n = 407

**Table 2. Constructs and items.**

| Constructs | Items | Sources |
|---|---|---|
| Information Quality (IQ) | The reviews on Taobao are complete. | Park, Lee, and Han [81] |
| | The reviews on Taobao are accurate. | |
| | The reviews on Taobao are objective. | |
| Source Credibility (SC) | People who left comments on Taobao are knowledgeable on their comment topics. | Sussman and Siegal [32] |
| | People who left comments on Taobao are experts on their comment topics. | |
| | People who left comments on Taobao are trustworthy. | |
| | People who left comments on Taobao are reliable. | |
| Information Usefulness (IU) | The reviews on Taobao are valuable. | Sussman and Siegal [32] |
| | The reviews on Taobao are informative. | |
| | The reviews on Taobao are helpful. | |
| Information Adoption (IA) | I will ask other consumers on Taobao to provide me with their suggestions before I go shopping. | Liang, Ho, Li, and Turban [80] |
| | I will consider the shopping experiences of other consumers on Taobao when I want to shop. | |
| | I am willing to buy the products recommended buy other consumers on Taobao. | |
| Personalized service (CS) | The products and services on Taobao are personalized to my needs. | Bock, Mangus, and Folse [77]; Srinivasan, Anderson, and Ponnavolu [82] |
| | Taobao makes purchase recommendations that match my needs. | |
| | Taobao enables me to order products that are tailor-made for me. | |
| | The advertisements and promotions that Taobao sends to me are tailored to my situation. | |
| | Taobao makes me feel that I am a unique customer. | |
| | Taobao provides many functions of online service (e.g., live chat). | |
| | I believe that Taobao is personalized to my needs. | |
| Habitual Behavior (HB) | Shopping at Taobao is something I do frequently. | Gupta, Su, and Walter [79] |
| | Shopping at Taobao is nature to me. | |
| | Shopping at Taobao is something I do without thinking. | |
| | I usually shop at Taobao because it is what I have always done. | |
| Perceived Switching Risk (PSR) | Switching to another online retailer would involve much hassle. | Gefen [78] |
| | Switching to another online retailer would take much time and effort. | |
| | Switching to another online retailer would cause many problems. | |
| | Switching to another online retailer would require much learning. | |
| Purchase Intention (PI) | If I need a product, I intend to purchase it on Taobao. | Venkatesh et al., [83] |
| | I intend to continue purchasing from Taobao in the future. | |
| | I am decided to shop on Taobao next time. | |
| | I will regularly use Taobao in the future. | |

divided into two parts. The first part included several questions on the demographics of the respondents. The second part comprised the measures of all constructs of our research model. All scales were adapted from prior research [32, 77–83]. Table 2 lists the questionnaire items used to measure each construct. A seven-point Likert scale anchored from 1 (completely disagree) to 7 (completely agree) was used to measure the model's variables.

### 3.3. Control variables

This study has contained four control variables: gender, education, age, and online shopping frequency. The participants of this study are all undergraduate students who have purchased from Taobao.com before.

Table 3. Means, standard deviations, and Pearson correlations.

| Constructs | Mean | S.D. | 1 | 2 | 3 | 4 | 5 | 6 | 7 | 8 |
|---|---|---|---|---|---|---|---|---|---|---|
| 1. Information quality | 4.31 | 0.86 | **.83** | | | | | | | |
| 2. Source credibility | 4.00 | 0.96 | .38** | **.87** | | | | | | |
| 3. Information usefulness | 4.39 | 0.97 | .35** | .41** | **.86** | | | | | |
| 4. Information adoption | 4.46 | 0.97 | .53** | .50** | .48** | **.72** | | | | |
| 5. Personalized service | 4.99 | 0.62 | −.02 | −.05 | .06 | .06 | **.74** | | | |
| 6. Habitual behavior | 5.86 | 0.92 | .01 | .03 | .04 | .06 | .47** | **.79** | | |
| 7. Perceived switching risk | 5.31 | 0.87 | .01 | −.04 | .04 | −.01 | .38** | .52** | **.81** | |
| 8. Purchase intention | 5.35 | 1.03 | .34** | .35** | .49** | .52** | .10** | .04 | −.04 | **.87** |

n = 407

square roots of AVE calculated for each of the constructs are along the diagonal.

** Correlation is significant at the .01 level (two-tailed).

## 4. Empirical results

### 4.1. Descriptive statistics

Table 3 shows the means, standard deviations, and Pearson correlations for all variables in this study. The table indicates that information quality is significantly and positively associated with information usefulness (r = 0.35, P<0.01). Source credibility is also significantly and positively associated with information usefulness (r = 0.41, P<0.01). Information usefulness is significantly and positively associated with information adoption (r = 0.48, P<0.01). Information adoption is significantly and positively associated with purchase intention (r = 0.52, P<0.01). Furthermore, personalized service is significantly and positively associated with purchase intention (r = 0.10, P<0.01). Personalized service is also significantly and positively associated with habitual behavior (r = 0.47, P<0.01) and perceived switching risk (r = 0.38, P<0.01). However, perceived switching risk (r = −0.04, n.s.) and habitual behavior (r = 0.04, n.s.) are not significantly associated with purchase intention.

### 4.2. Measurement model

We perform a confirmatory factor analysis to test whether the data fit the hypothesized measurement model. Table 4 shows the results of the goodness-of-fit indices for the proposed research model. The hypothesized model shows satisfactory goodness-of-fit indices: χ2/d.f. ratio is approximately 2.67, GFI = 0.91, CFI = 0.92, NFI = 0.91, TLI = 0.91, and RMSEA = 0.06. These results show that the goodness-of-fit measures for the hypothesized model meet the requirements of the benchmark fit indices (χ2/d.f.<3, GFI>0.90, CFI>0.90, NFI>0.90, TLI>0.90, and RMSEA<0.08) [84, 85], thereby indicating that the conceptual model fits the data reasonably well.

To test the convergent and discriminant validity of the measures, composite reliability (CR), average variance extracted (AVE), and square root of average, and variance extracted ($\sqrt{}$AVE) are examined in this study. Convergent validity is acceptable if the value of CR

Table 4. Goodness of fit.

| Constructs / model | $\chi^2$/d.f. | P-value | GFI | CFI | NFI | TLI | RMSEA |
|---|---|---|---|---|---|---|---|
| *Thresholds* | < 3 | > .05 | > .90 | > .90 | > .90 | > .90 | < .08 |
| Hypothesized Model | 1260.03/472 | 0.000 | 0.91 | 0.92 | 0.91 | 0.91 | 0.06 |

n = 407

**Table 5. Confirmatory factor analysis results.**

| Construct | Item | Standardized estimates | CR | AVE | √AVE | Cronbach's α |
|---|---|---|---|---|---|---|
| IQ | IQ1 | 0.75 | 0.87 | 0.69 | 0.83 | 0.87 |
|  | IQ2 | 0.88 |  |  |  |  |
|  | IQ3 | 0.86 |  |  |  |  |
| SC | SC1 | 0.82 | 0.93 | 0.76 | 0.87 | 0.93 |
|  | SC2 | 0.84 |  |  |  |  |
|  | SC3 | 0.91 |  |  |  |  |
|  | SC4 | 0.91 |  |  |  |  |
| IU | IU1 | 0.91 | 0.90 | 0.74 | 0.86 | 0.89 |
|  | IU2 | 0.73 |  |  |  |  |
|  | IU3 | 0.93 |  |  |  |  |
| IA | IA1 | 0.80 | 0.76 | 0.52 | 0.72 | 0.76 |
|  | IA2 | 0.65 |  |  |  |  |
|  | IA3 | 0.71 |  |  |  |  |
| PS | PS1 | 0.79 | 0.89 | 0.54 | 0.74 | 0.80 |
|  | PS2 | 0.82 |  |  |  |  |
|  | PS3 | 0.74 |  |  |  |  |
|  | PS4 | 0.54 |  |  |  |  |
|  | PS5 | 0.79 |  |  |  |  |
|  | PS6 | 0.73 |  |  |  |  |
|  | PS7 | 0.70 |  |  |  |  |
| HB | HB1 | 0.86 | 0.87 | 0.63 | 0.79 | 0.88 |
|  | HB2 | 0.86 |  |  |  |  |
|  | HB3 | 0.77 |  |  |  |  |
|  | HB4 | 0.67 |  |  |  |  |
| PSR | PSR1 | 0.91 | 0.88 | 0.65 | 0.81 | 0.89 |
|  | PSR2 | 0.85 |  |  |  |  |
|  | PSR3 | 0.71 |  |  |  |  |
|  | PSR4 | 0.74 |  |  |  |  |
| PI | PI1 | 0.73 | 0.92 | 0.75 | 0.87 | 0.87 |
|  | PI2 | 0.87 |  |  |  |  |
|  | PI3 | 0.96 |  |  |  |  |
|  | PI4 | 0.89 |  |  |  |  |

n = 407, IQ = information quality, SC = source credibility, IU = information usefulness, IA = information adoption, PS = personalized service, HB = habitual behavior, PSR = perceived switching risk, PI = purchase intention.

exceeds 0.70 and the value of AVE is above 0.50 [84, 85]. Table 5 shows that the CR and AVE values of all constructs meet these requirements. Thus, the convergent validity of the measures is sufficient in this study. Furthermore, discriminant validity is satisfactory if the square root of AVE is greater than the off-diagonal elements in the corresponding rows and columns of the Pearson correlation matrix [82]. Table 3 shows that the square roots of AVE on the main diagonal are greater than those of the off-diagonal elements in the corresponding rows and columns of the matrix. Thus, the results evidently indicate good discriminant validity of the measure in this study.

Internal consistency reliability is also tested for each construct of the proposed model. Cronbach's alpha for information quality, source credibility, information usefulness, information adoption, personalized service, habitual behavior, perceived switching risk, and purchase

**Table 6. Harmar's one-factor test.**

| Constructs / model | $\chi^2/d.f.$ | P-value | GFI | CFI | NFI | TLI | RMSEA |
|---|---|---|---|---|---|---|---|
| *Thresholds* | *< 3* | *> .05* | *> .90* | *> .90* | *> .90* | *> .90* | *< .08* |
| One factor model | 6501.17/495 | 0.000 | 0.37 | 0.41 | 0.39 | 0.37 | 0.17 |

n = 407

intention are 0.87, 0.93, 0.89, 0.76, 0.80, 0.88, 0.89, and 0.87, respectively. These values are the suggested criteria of 0.60 [84, 85]. Thus, the results indicate good reliability of the measurement scales.

Based on the self-reported survey, common method bias is examined in this study. According to Podsakoff, MacKenzie, and Lee [86], common method variance will appear if the results of principle component analysis show a single factor from unrotated factor solution or if a first factor explains the majority of the variance. We enter all items into principle component analysis and examine the unrotated factor solution. Eight factors emerge with an eigenvalue greater than 1.0, which accounts for 70.36% of variance. The first factor accounts for 9.12% of variance. The results indicate that neither a single factor nor the first factor explains the majority of the variance. Furthermore, we perform a CFA of the one-factor model. The results of this one-factor model show a poor model fit ($\chi 2/df = 13.13$, GFI = 0.37, CFI = 0.41, NFI = 0.39, TLI = 0.37, and RMSEA = 0.17) (see Table 6). Thus, common method variance does not seem to be a serious problem in our sample data.

## 4.3. Structural model

Structural equation model is used to test the hypotheses of our research model. The results in Fig 2 show that source credibility is significantly and positively related to information usefulness ($\beta = 0.621$, P<0.001), providing support for Hypothesis H1. Information quality is also significantly and positively related to information usefulness ($\beta = 0.216$, P<0.001), providing support for Hypothesis H2. Information usefulness is significantly and positively related to information adoption ($\beta = 0.618$, P<0.001), providing support for Hypothesis H3. Information adoption is significantly and positively related to online purchase intention ($\beta = 0.481$, P<0.001), providing support for Hypothesis H4. Furthermore, personalized service is significantly and positively related to online purchase intention ($\beta = 0.100$, P<0.05), providing support for Hypothesis H5. Personalized service is also significantly and positively related to habitual behavior ($\beta = 0.509$, P<0.001) and perceived switching risk ($\beta = 0.508$, P<0.001), providing support for Hypotheses H6 and H7. Moreover, perceived switching risk is significantly and positively related to online purchase intention ($\beta = 0.125$, P<0.05), providing support for Hypothesis H8. Finally, habitual behavior is significantly and negatively related to online purchase intention ($\beta = -0.212$, P<0.001). Thus, Hypothesis H9 is not supported.

## 5. Discussion

The findings of this research offer novel and important implications for academic researchers and business managers of e-commerce. First, prior studies have ignored the online consumer behavior of the post-90s Chinese generation. Most research has investigated Chinese generational cohorts in psychology and organizational management literature [17]. This study extends generational cohort theory by examining the online shopping behavior of the post-90s Chinese generation. As far as we know, this study is one of the first to investigate this generational cohort in the marketing literature.

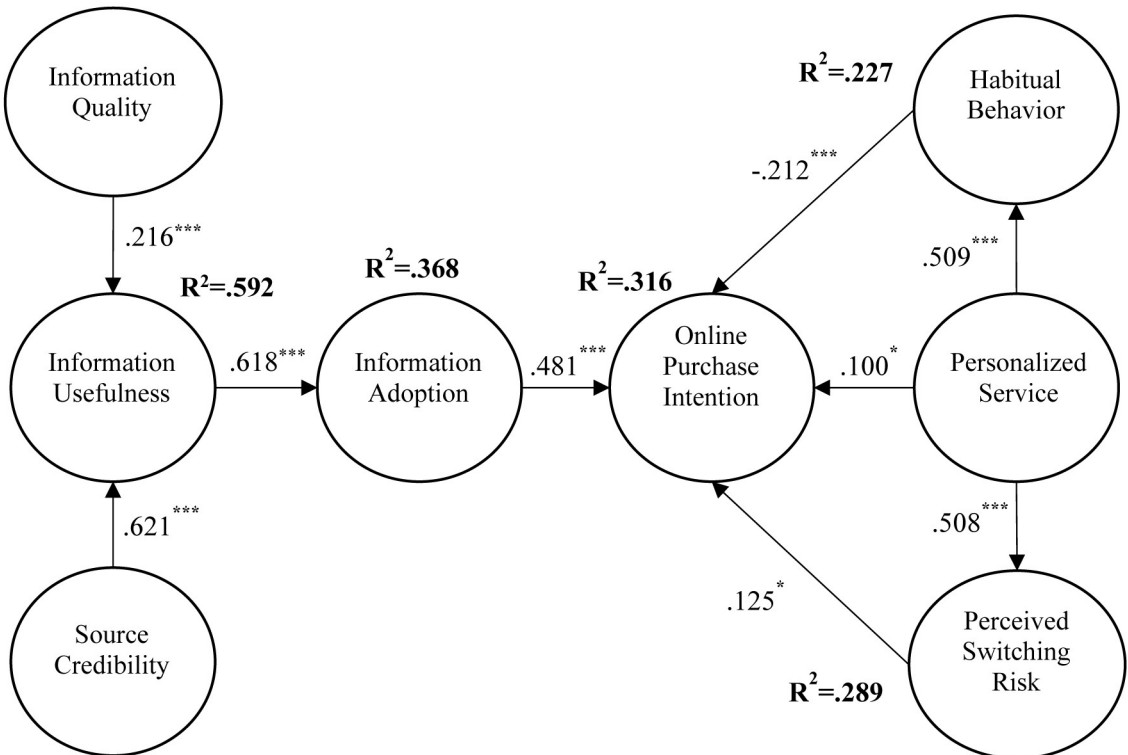

Figure 2. The path coefficient of full model
(***P<0.001, **P<0.01, *P<0.05)

Fig 2. The path coefficient of full model. ***P<0.001, **P<0.01, *P<0.05.

Second, this study enriches our understanding of the influence of online information adoption. The findings explain the impact of informational factors on the consumer decision-making process. Source credibility and information quality are two key factors that determine the usefulness of information [11]. Consumers often perceive that information and comments on shopping websites from credible people are beneficial for their purchase decisions [41]. Consumers also believe that obtaining complete and accurate information is useful to evaluate products and services [32]. Furthermore, consumers tend to adopt useful information to purchase products online [39]. Adoption of useful information reduces information asymmetry and costs related to consumer decisions [46]. Our findings provide evidence to expand our knowledge on information adoption theory and TAM in the online shopping context.

Third, this study examines the influence of personalized service in the online shopping environment. Research shows that consumers are more likely to purchase online when online retailers provide personalized services [53]. Such likelihood exists because personalized services make online shopping more interesting, more hedonic, and more economical for consumers [54]. In this study, we extend prior research in two specific areas. On the one hand, we show that personalization may lead to consumer habitual behavior because consumers repeatedly obtain high values from the personalization process, creating consumer dependence and

attachment to online retailers [54]. On the other hand, we demonstrate that personalized services make switching to new retailers harder for consumers because personalization raises consumer perceptions of switching risk [59] and ensures lower level of uncertainty [53].

In addition, this study provides new perspective on the impact of perceived switching risk on online purchase intention. According to protection motivation theory [62, 63], consumers often face higher risks when switching to new retailers because they are uncertain about the new retailers [68]. One effective strategy for consumers is to purchase at current online retailers because they already know them [43]. This study extends our understanding of protection motivation theory in the online shopping environment.

Finally, this study sheds new light on the role of habitual behavior in the consumer decision-making process in the online context. Instrumental learning theory explains that consumers may obtain great benefits and values when they repeatedly purchase online [76]. If this process repeats over time, consumer's purchasing behavior will be strengthened and will become habitual [71]. However, our findings do not support instrumental learning theory. By contrast, habitual behavior is negatively related to online purchase intention. One possible reason is that the post-90s Chinese generation tends to engage in novel-seeking behavior. They seek information, compare different websites, and do not rely on a specific online store to purchase products. Furthermore, according to two-factor theory, two separate psychological processes operate when consumers repeatedly shop at a specific online store [56]. On the one hand, repeated purchase increases consumer familiarity with the retailers. On the other hand, repeated purchase may lead to boredom when consumers shop at the same retailer over time [56]. Thus, habitual behavior probably negatively affects online purchase intention due to consumers' boredom when they repeatedly shop online.

Findings of this study also provide implications for business practitioners. The post-90s Chinese generation is the largest user group on the Internet. Growing up with the Internet and digital technology, this generation comprise the majority of online shoppers. Given this generation's unique values, attitudes, behaviors, and their great purchasing potential, business managers should understand and provide marketing strategies that fit their preferences. Furthermore, the findings of this study suggest the important role of informational factors. Business managers should encourage credible sources of information to post their comments and recommendations about products and services. Business managers should also collect and disseminate high-quality information on their websites.

In addition, personalized service has become an important attribute for online businesses. Online retailers that provide interactive, customized, and more responsive services can easily satisfy consumer needs. Moreover, personalized service raises consumers' perceptions of risk when they intend to switch to other retailers. Personalization also enhances consumer dependence and attachment to current retailers. Thus, business managers should design and offer personalized strategies (e.g., personalized webpage, live chat, customized information, personal recommendations, and suggestions) to enhance their service quality for consumers.

In addition, perceived switching risk increases consumer purchase intention on current online stores. Business managers should use marketing strategies to raise consumer perceptions of risk when they switch to new retailers. Business managers should also persuade consumers through additional benefits, improved safety, and fewer costs. However, business managers should pay attention to consumer habitual behavior. Consumers may not repeatedly purchase at current stores due to their feelings of boredom. Thus, online retailers should use strategies to create novel experiences for consumers. For example, online retailers may change their website interface, provide new ads, launch different promotion activities, or create hedonic activities to attract and retain consumers.

## 6. Conclusions

This study aims to propose an integrated model that determines the effects of information adoption, personalized service, perceived switching risk, and habitual behavior of the online purchase intention of the post-90s Chinese generation. Findings of this study contribute to the knowledge of current literature in which it extends the generalizability of generational cohort theory in emerging economy (i.e. China) and in the fields of marketing and consumer behavior. Furthermore, findings of this study also provide implications for business managers making better decisions that provide products and services for younger consumers in China market.

Although this study has been conducted rigorously, we acknowledge a few limitations that should be addressed in future research. Data collected for the independent and dependent variables are based on self-reports, which may have biased our results due to the social desirability problem [86]. To overcome this common method bias, future research should obtain more objective and potentially less biased measures and use more advanced methods to deal with common method bias. Furthermore, our sample data are collected from college students. This may affect the generalization of this study because not all members of the post-90s generation are college students. Because population of this study was difficult to identify and sample frame was not available, selection of college students may be the most suitable sample for this study. However, this convenient sampling technique may affect the representatives and results of the analysis. Future research should use random sampling techniques to collect data from different generational cohorts and from different cultural environments (e.g., Southeast Asia, Europe, and so on). In addition, many psychological factors may affect consumers' attitudes and behavior in online shopping environment. Future research should deal with other variables such as emotion, persuasion, perceptions, etc.

## Author Contributions

**Conceptualization:** Van Thac Dang.

**Data curation:** Thinh Truong Vu.

**Investigation:** Thinh Truong Vu.

**Methodology:** Jianming Wang.

**Supervision:** Van Thac Dang.

**Validation:** Thinh Truong Vu.

**Writing – original draft:** Van Thac Dang.

**Writing – review & editing:** Jianming Wang.

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
