## [Decision Letter · Decision Letter 0]

17 Jan 2020

PONE-D-19-28617

An integrated model of younger generation's online shopping behavior based on empirical evidence gathered from an emerging economy

PLOS ONE

Dear Professor Wang,

Thank you for submitting your manuscript to PLOS ONE. After careful consideration, we feel that it has merit but does not fully meet PLOS ONE’s publication criteria as it currently stands. Therefore, we invite you to submit a revised version of the manuscript that addresses the points raised during the review process.

We would appreciate receiving your revised manuscript by Mar 02 2020 11:59PM. To enhance the reproducibility of your results, we recommend that if applicable you deposit your laboratory protocols in protocols.io, where a protocol can be assigned its own identifier (DOI) such that it can be cited independently in the future. For instructions see: http://journals.plos.org/plosone/s/submission-guidelines#loc-laboratory-protocols

We look forward to receiving your revised manuscript.

Kind regards,

Baogui Xin, Ph.D.

Academic Editor

PLOS ONE

Journal Requirements:

https://www.emerald.com/insight/content/doi/10.1108/APJML-01-2018-0038/full/html

In your revision ensure you cite all your sources (including your own works), and quote or rephrase any duplicated text outside the methods section. Further consideration is dependent on these concerns being addressed.

4. Thank you for stating the following financial disclosure: "No"

Please provide an amended Funding Statement that declares *all* the funding or sources of support received during this specific study (whether external or internal to your organization) as detailed online in our guide for authors at http://journals.plos.org/plosone/s/submit-now.  

Please state what role the funders took in the study.  If any authors received a salary from any of your funders, please state which authors and which funder. If the funders had no role, please state: "The funders had no role in study design, data collection and analysis, decision to publish, or preparation of the manuscript."

5. Thank you for stating the following in your Competing Interests section: "No"

Reviewers' comments:

Reviewer's Responses to Questions

**Comments to the Author**

1. Is the manuscript technically sound, and do the data support the conclusions?

Reviewer #1: Yes

Reviewer #2: Yes

2. Has the statistical analysis been performed appropriately and rigorously? 

Reviewer #1: Yes

Reviewer #2: Yes

3. Have the authors made all data underlying the findings in their manuscript fully available?

Reviewer #1: Yes

Reviewer #2: No

4. Is the manuscript presented in an intelligible fashion and written in standard English?

Reviewer #1: Yes

Reviewer #2: Yes

5. Review Comments to the Author

Reviewer #1: Overview and general recommendation:

Drawing upon literature of Generational cohort theory and Information adoption in online shopping, the authors claim to investigate younger generation's online shopping behavior. The authors construct a research model to investigate the effects of information adoption, personalized service, perceived switching risk, and habitual behavior on purchase intention in online shopping environment. Using the empirical data, the study suggests that information adoption, personalized service, and perceived switching risk are the most important predictors of online purchase intention. Overall, the topic seems interesting. However, the motivation, research gap, and theoretical foundations should be clearly clarified based on a thorough literature review. In addition, the representative of sample data should be revised or justified. Following are specific comments for further improving this work.

Major comments:

Introduction

Firstly, this study indicated that the gap between China’s new and older generations is arguably wider. Whereas, the research motivation is still not clearly identified. Firstly, the authors should explain why the “research gap” is important from a more theoretical perspective. Only relying on the practical and managerial importance is not a strong motivation. Secondly, how the prior studies investigate this “gap”? To conclude, the authors should clearly identify the research gap and the research motivation based on solid literature review.

Literature

The literature review should be extended and integrated to include significant missing research domains. For example, the study seems lack a theoretical foundation. The authors briefly discuss the “information adoption in online shopping”. Whereas, “information adoption in online shopping” is a very broad field, many theories are applied in this field, for example the theories from social psychology, such as the TRA as the authors mentioned, and numerous research models from the school of TAM, such as TAM1,2,3 UTAUT1.2. Consequently, the author may lose its focus by only using the very broad “beliefs � behavioral intention” linkage as its “theoretical foundation”. The authors should provide a clearer arguments and a solid literature review regarding this issue.

In addition, the authors developed a rather complicated mode having 8 constructs. But it seems difficult to find a theoretical support of why these constructs can be included in a research model. As a result, the study also fails to explain why other constructs are not included. Hence, we cannot clearly identify what theoretical contributions (not a specific construct) are made on what theory. I strongly suggest the authors to provide a missing literature review in this section

Method

1. Regarding the data sample, how do you determine the reasonable percentages of the respondents to ensure the representativeness of the lager population? For example, 1). 71.5% females are females. 2). About 96.6% of the respondents purchased online more than three times a month, and only 3.4% of the respondents purchased online less than three times a month.

2. In the research process:

2.1 Did the authors conducted pretest to ensure the content validity of the questionnaire?

2.2 How did the author deal with the Chinese-English translation of the questionnaire to ensure the quality of it?

Discussion and Conclusion

In the section of “theoretical implication” section: Instead of writing “theoretical implication”, the authors actually seem write the “Discussion” or “Conclusion” section, by explaining the new findings, how the results are in accordance with prior studies. Hence, I suggest the authors to revise the secondary heading to “discussion”.

And importantly, the authors should add one more section to present their “theoretical implication”, and discuss how this study can contribute to which theory, or which area in which way?

Reviewer #2: This study draws from generational cohort theory and examines examines online shopping behavior of the post-90s Chinese generation. The introduction gives a good presentation of the topic and problem. To improve the introduction, and highlight the contribution and novelty, some changes and revisions are needed. After introducing the problem, then discuss what some best studies in the area have done and what they have missed (research gap). Then, explain why we need to address this gap, which should be beyond the fact that no studies have examined post-90s Chinese generation. Here you need to explain why this is important to be examined (see comment below about section 2). The describe how you will address this gap and present your major findings which would make for your contribution. Also, try to keep the paragraphs at an equal size as it makes it easier to follow. For example the first is very short while the fourth is rather long.

More details are needed what needs, motivations, characteristics etc, the post-90s Chinese generation has and how Generational cohort theory affects that. It is mentioned in the end of section 2.1 that 2.2 “discusses the influential factors that affect the online purchasing behavior of the post-90s Chinese generation.” however, the arguments in 2.2 are not focused on the post-90s Chinese generation. A new section here could help to address the issue.

The method has been performed properly.

For the common method bias, Podsakoff et al (2003) actually suggest to avoid Harman’s one-factor test and propose other ones. However, there have been recent papers that deal with this issue and argue that Harman’s one factor test is appropriate. Please update this.

The findings from SEM do not mention the r-square values. This is critical information to assess the model.

As the authors have a large number of participants, you may try looking for moderating effects from the control variables (e.g., age?), or you have more details on the online shopping frequency. See some related work in the end.

Since the study highlights as its main contribution the fact that the sample is born in the 90s, more details are needed about it. Almost all of the sample (except 2.5%) is under 24, with the vast majority being under 20. Also, all participants are students. What about participants that are older than 24 but still born in the 90s? (This is very briefly mentioned in the limitations; how can you overcome this?) What is the distribution for those under 20? Maybe using year of birth instead of age would clarify such issues.

In 5.1 it is mentioned that one of the theoretical contributions is the extension of generational cohort theory because the sample was born in the 90s. It is not explained at all what the findings imply for the theory and how the study contributes to it. This is directly related with the discussion on generational cohort theory in 2.1 and around the post-90s Chinese generation that is rather limited in the background section.

Future research can be expanded as it is very limited now. The related work below can also help towards that direction.

Pappas, I. O. (2018). User experience in personalized online shopping: a fuzzy-set analysis. European Journal of Marketing, 52(7/8), 1679-1703.

Pappas, I. O., Kourouthanassis, P. E., Giannakos, M. N., & Chrissikopoulos, V. (2017). Sense and sensibility in personalized e‐commerce: How emotions rebalance the purchase intentions of persuaded customers. Psychology & Marketing, 34(10), 972-986.

Huang, J., & Zhou, L. (2019). The dual roles of web personalization on consumer decision quality in online shopping. Internet Research.

6. PLOS authors have the option to publish the peer review history of their article (what does this mean?). If published, this will include your full peer review and any attached files.

Reviewer #1: No

Reviewer #2: No

---

## [Author Response · Author response to Decision Letter 0]

3 Apr 2020

Dear Editor:

Thank you for giving us a chance to revise our manuscript entitled "An integrated model of the younger generation’s online shopping behavior based on empirical evidence gathered from an emerging economy") for PLOS ONE.

 The comments from you and reviewers are very helpful. We have considered carefully the comments and revised our paper. Our point-to-point responses are shown in the following pages. We wish you would agree with us that we have addressed all the concerns in this revision.

Best regards,

Authors of the Manuscript 

Reviewer #1:

Dear professor: Many thanks for your insightful and detailed comments. We have endeavored to address all of your comments in a very systematic and comprehensive manner. This has greatly improved the academic rigor and readability of our manuscript, and for this we are extremely grateful to you. Please kindly see our responses as follows

Introduction

Comment 1: Firstly, this study indicated that the gap between China’s new and older generations is arguably wider. Whereas, the research motivation is still not clearly identified. Firstly, the authors should explain why the “research gap” is important from a more theoretical perspective. Only relying on the practical and managerial importance is not a strong motivation. Secondly, how the prior studies investigate this “gap”? To conclude, the authors should clearly identify the research gap and the research motivation based on solid literature review.

Our reply: We have revised the introduction section. We added a paragraph to explain the motivation and contribution of our research to theory and practice.

“…Generational difference has become an important gap since social value changed through generation replacement in China (Howe and Strauss, 2000). Transitional economy like China has witnessed a wider gap between generations after its social reform (Liao and Zhang, 2007). Although generational cohort theory is universally applicable in Western societies (Bilgihan, 2016), an investigation of generational value differences in China would shed a new light on the generalization of the theory. Furthermore, prior researchers tend to apply generational cohort theory in the fields of organizational behavior and human resource management (Liu, 2011; Tang et al., 2017). The absence of its application in marketing management limits the generalizability of the theory. Investigating online shopping behavior of Chinese younger generation would extend generational cohort theory in the field of marketing and consumer behavior (i.e. e-commerce). In addition, a lack of empirical evidence on consumer behavior of Chinese younger generation has provided very limited knowledge to business managers’ decision making in China market. Findings of this study will contribute to enhance quality of business policy for Internet commerce in China. Therefore, this study draws on the theoretical foundation of generational cohort theory to investigate important factors that affect the online shopping behavior of the post-90s Chinese generation….” (Page. 3-4).

Literature

Comment 2: The literature review should be extended and integrated to include significant missing research domains. For example, the study seems lack a theoretical foundation. The authors briefly discuss the “information adoption in online shopping”. Whereas, “information adoption in online shopping” is a very broad field, many theories are applied in this field, for example the theories from social psychology, such as the TRA as the authors mentioned, and numerous research models from the school of TAM, such as TAM1,2,3 UTAUT1.2. Consequently, the author may lose its focus by only using the very broad “beliefs � behavioral intention” linkage as its “theoretical foundation”. The authors should provide a clearer argument and a solid literature review regarding this issue.

Our reply: Thank you for your suggestion. We have revised the literature review section. We added section 2.1. to explain the generation cohort theory and Chinese generational cohort. We also added section 2.2. to discuss information adoption theory and technology acceptance model. These theories are used as theoretical basis for the hypothesis development in our research.

Comment 3: In addition, the authors developed a rather complicated mode having 8 constructs. But it seems difficult to find a theoretical support of why these constructs can be included in a research model. As a result, the study also fails to explain why other constructs are not included. Hence, we cannot clearly identify what theoretical contributions (not a specific construct) are made on what theory. I strongly suggest the authors to provide a missing literature review in this section.

Our reply: We have proposed a complicated research model which includes several constructs. In fact, it’s very difficult to find a single theory to support the whole model and develop relationships between variables based on a single theory. Thus, we used different theories including generational cohort theory, theory of information adoption and technology acceptance model, behavioral learning theory, and protection motivation theory, etc., to infer relationships between variables. A failure of using a single theory to support the whole model is a major limitation in our research. We will try to overcome this problem in future research. Thank you!

Method

Comment 4: Regarding the data sample, how do you determine the reasonable percentages of the respondents to ensure the representativeness of the lager population? For example, 1). 71.5% females are females. 2). About 96.6% of the respondents purchased online more than three times a month, and only 3.4% of the respondents purchased online less than three times a month.

Our reply: We acknowledge that this is another limitation of our study. We could not overcome this limitation because it was very difficult to identify the population and obtain a sample frame for our study. Therefore, we selected college students as our target sample. We have revised and discussed more details about this limitation in the conclusions section. 

“…Furthermore, our sample data are collected from college students. This may affect the generalization of this study because not all members of the post-90s generation are college students. Because population of this study was difficult to identify and sample frame was not available, selection of college students may be the most suitable sample for this study. However, this convenient sampling technique may affect the representatives and results of the analysis. Future research should use random sampling techniques to collect data from different generational cohorts and from different cultural environments (e.g., Southeast Asia, Europe, and so on)….” (Page. 20).

Comment 5: Did the authors conducted pretest to ensure the content validity of the questionnaire? How did the author deal with the Chinese-English translation of the questionnaire to ensure the quality of it?

Our reply: We have revised this section and discussed in more details.

“A preliminary version of the questionnaire was formulated using forward and backward translation from English to Chinese, and vice versa. Two professional translators who are proficient in both English and Chinese were recruited to perform and ensure the forward and backward translation. A pilot test was conducted with 26 students in the author’s university to check the appropriateness of the wording and meaning of items in the questionnaire….” (Page. 12).

Discussion and Conclusion

Comment 6: In the section of “theoretical implication” section: Instead of writing “theoretical implication”, the authors actually seem write the “Discussion” or “Conclusion” section, by explaining the new findings, how the results are in accordance with prior studies. Hence, I suggest the authors to revise the secondary heading to “discussion”. And importantly, the authors should add one more section to present their “theoretical implication”, and discuss how this study can contribute to which theory, or which area in which way?

Our reply: Thank you for your suggestion. We have revised the “conclusion and discussions” section. We discussed theoretical and practical implications and limitations in more details. 

Reviewer #2: 

Dear professor: Many thanks for your insightful and detailed comments. We have endeavored to address all of your comments in a very systematic and comprehensive manner. This has greatly improved the academic rigor and readability of our manuscript, and for this we are extremely grateful to you. Please kindly see our responses as follows

Comment 1: This study draws from generational cohort theory and examines online shopping behavior of the post-90s Chinese generation. The introduction gives a good presentation of the topic and problem. To improve the introduction, and highlight the contribution and novelty, some changes and revisions are needed. After introducing the problem, then discuss what some best studies in the area have done and what they have missed (research gap). Then, explain why we need to address this gap, which should be beyond the fact that no studies have examined post-90s Chinese generation. Here you need to explain why this is important to be examined (see comment below about section 2). The describe how you will address this gap and present your major findings which would make for your contribution. Also, try to keep the paragraphs at an equal size as it makes it easier to follow. For example, the first is very short while the fourth is rather long.

Our reply: Many thanks for your suggestions! We have revised our paper and added many paragraphs to make the paper better. In the introduction section, we added one paragraph to explain the motivation and contribution of our research. In the literature review section, we added section 2.1. to explain generational cohort theory and Chinese generational cohort. We also added section 2.2. to discuss theory of information adoption and technology acceptance model. These will help to provide theoretical basis for the hypothesis development in our research. Furthermore, we have also revised the paper so that paragraphs appear in a more balanced manner. (Page. 5-6).

Comment 2: More details are needed what needs, motivations, characteristics etc, the post-90s Chinese generation has and how Generational cohort theory affects that. It is mentioned in the end of section 2.1 that 2.2 “discusses the influential factors that affect the online purchasing behavior of the post-90s Chinese generation.” however, the arguments in 2.2 are not focused on the post-90s Chinese generation. A new section here could help to address the issue. 

Our reply: Thank you. We have added section 2.2. to discuss more details about the characteristics of the post-90s Chinese generation which is the main research sample of our study. (Page. 6).

Comment 3: The method has been performed properly.

Our reply: Thank you!

Comment 4: For the common method bias, Podsakoff et al (2003) actually suggest to avoid Harman’s one-factor test and propose other ones. However, there have been recent papers that deal with this issue and argue that Harman’s one factor test is appropriate. Please update this.

Our reply: Thank you for your suggestion. We acknowledge that Harman’s one-factor model is not the most suitable method to deal with common method bias. However, with our current knowledge, we used principle factor analysis with unrotated solution and CFA with one-factor model to test the common method bias. An instrumental variable method should be more suitable, but we have not used this when we designed and selected data. We will be careful and use different methods in future research.

Comment 5: The findings from SEM do not mention the r-square values. This is critical information to assess the model.

Our reply. We have added the R-square values in the model.

Comment 6: As the authors have a large number of participants, you may try looking for moderating effects from the control variables (e.g., age?), or you have more details on the online shopping frequency. See some related work in the end.

Our reply: Thank you for your suggestions. I would be more interesting to determine some moderating variables in this study. However, considering the focus of this study and avoiding too much variables that may make confusion in this study, we only focused on relevant variables as presented in our current research model. We will deal with more variables and the moderating effect in future research. Thank you!

Comment 7: Since the study highlights as its main contribution the fact that the sample is born in the 90s, more details are needed about it. Almost all of the sample (except 2.5%) is under 24, with the vast majority being under 20. Also, all participants are students. What about participants that are older than 24 but still born in the 90s? (This is very briefly mentioned in the limitations; how can you overcome this?) What is the distribution for those under 20? Maybe using year of birth instead of age would clarify such issues.

Our reply: We acknowledge that this is a major limitation of our study. We could not overcome this limitation because it was very difficult to identify the population and obtain a sample frame for our study. Therefore, we selected college students as our target sample. We have revised and discussed more details about this limitation in the conclusions section. 

“…Furthermore, our sample data are collected from college students. This may affect the generalization of this study because not all members of the post-90s generation are college students. Because population of this study was difficult to identify and sample frame was not available, selection of college students may be the most suitable sample for this study. However, this convenient sampling technique may affect the representatives and results of the analysis. Future research should use random sampling techniques to collect data from different generational cohorts and from different cultural environments (e.g., Southeast Asia, Europe, and so on)….” (Page. 20).

Comment 8: In 5.1 it is mentioned that one of the theoretical contributions is the extension of generational cohort theory because the sample was born in the 90s. It is not explained at all what the findings imply for the theory and how the study contributes to it. This is directly related with the discussion on generational cohort theory in 2.1 and around the post-90s Chinese generation that is rather limited in the background section.

Our reply: We have revised the introduction and discussion and conclusions sections. We discussed in more details the contribution to both theory and practice. (Page. 3-4 and 17-19).

Comment 9: Future research can be expanded as it is very limited now. The related work below can also help towards that direction.

Our reply. Thank you for your suggestions. We have revised the limitation and discussed in more details.

“Although this study has been conducted rigorously, we acknowledge a few limitations that should be addressed in future research. Data collected for the independent and dependent variables are based on self-reports, which may have biased our results due to the social desirability problem (Podsakoff et al., 2003). To overcome this common method bias, future research should obtain more objective and potentially less biased measures and use more advanced methods to deal with common method bias. Furthermore, our sample data are collected from college students. This may affect the generalization of this study because not all members of the post-90s generation are college students. Because population of this study was difficult to identify and sample frame was not available, selection of college students may be the most suitable sample for this study. However, this convenient sampling technique may affect the representatives and results of the analysis. Future research should use random sampling techniques to collect data from different generational cohorts and from different cultural environments (e.g., Southeast Asia, Europe, and so on). In addition, many psychological factors may affect consumers’ attitudes and behavior in online shopping environment. Future research should deal with other variables such as emotion, persuasion, perceptions, etc.” (Page. 19-20).

---

## [Editor Report · Decision Letter 1]

10 Apr 2020

An integrated model of younger generation's online shopping behavior based on empirical evidence gathered from an emerging economy

PONE-D-19-28617R1

Dear Dr. Wang,

We are pleased to inform you that your manuscript has been judged scientifically suitable for publication and will be formally accepted for publication once it complies with all outstanding technical requirements.

With kind regards,

Baogui Xin, Ph.D.

Academic Editor

PLOS ONE
---

## [Editor Report · Acceptance letter]

23 Apr 2020

PONE-D-19-28617R1 

An integrated model of the younger generation’s online shopping behavior based on empirical evidence gathered from an emerging economy 

Dear Dr. Wang:

I am pleased to inform you that your manuscript has been deemed suitable for publication in PLOS ONE. Congratulations! Your manuscript is now with our production department. 

With kind regards,

on behalf of

Prof. Baogui Xin 

Academic Editor

PLOS ONE